# Shark discards in selective and mixed-species pelagic longline fisheries

**Gareth L. Jordaan** [1]*, **Jorge Santos**[2], **Johan C. Groeneveld**[1,3]

**1** Oceanographic Research Institute, Durban, South Africa, **2** Faculty of Bio-Sciences, Fisheries and Economics, UiT The Arctic University of Norway, Tromsø, Norway, **3** School of Life Sciences, University of KwaZulu-Natal, Pietermaritzburg, KwaZulu-Natal, South Africa

☯ These authors contributed equally to this work.
* jordaan.gareth@gmail.com

**Data Availability Statement:** All relevant data are within the manuscript and its Supporting Information files.

**Funding:** This work was funded by National Research Foundation (NRF) incentive fund (grant

## Abstract

The conservation status of several pelagic shark species is considered vulnerable with declining populations, yet data on shark fishing mortality remain limited for large ocean regions. Pelagic sharks are increasingly retained by mixed-species fisheries, or are discarded and not reported by selective fisheries for tunas (*Thunnus* spp.) or swordfish (*Xiphias gladius*). We estimated the fishing mortality of sharks (landings plus discard mortalities) in a South African-flagged pelagic longline fishery with diverse targeting and discard behaviour. A hierarchical cluster analysis was used to stratify the fleet according to the relative proportions of tunas, swordfish, blue sharks (*Prionace glauca*) and shortfin mako sharks (*Isurus oxyrinchus*) landed by individual vessels between 2013 and 2015. A spatial analysis of logbook data indicated that subfleets operated in distinct fishing areas, with overlap. Approximately 5% of all commercial longlines set during 2015 were sampled by a fisheries-independent observer, and the species, discard ratios and physical condition at discard of 6 019 captured sharks were recorded. Blue sharks and shortfin makos dominated observed shark catches, which were comprised of nine species and two species groups. Some 47% of observed sharks were retained and 20% were discarded in good physical condition. Only 4% of shortfin makos were discarded, compared to 68% of blue sharks. Blue shark discard mortality rates were twice as high as published at-vessel mortality rates, suggesting that onboard handling, among other factors, contributed to discard mortalities. Extrapolation to total fishing effort indicated a near 10-fold increase in blue shark and shortfin mako fishing mortality compared to an earlier study (1998–2005). Escalating shortfin mako fishing mortality was attributed to increased targeting to supply higher market demand. Discarding of blue sharks by selective fishing for tunas and swordfish had a greater impact on their fishing mortality than retention by shark-directed fleets. Higher levels of observer sampling are required to increase confidence in discard ratio estimates.

number 96309) to Johan Groeneveld, as well as through a college bursary provided by the University of KwaZulu-Natal (UKZN). The funders had no role in study design, data collection and analysis, decision to publish, or preparation of the manuscript.

**Competing interests:** The authors have declared that no competing interests exist.

## Introduction

Pelagic longline fisheries for tunas and swordfish typically have high incidental catches of sharks, of which most are discarded overboard as unwanted catch [1, 2]. Hooked sharks often die during capture or shortly thereafter as a result of physical injuries or physiological stress, and high post-discard mortality rates have been found for several pelagic shark species [3–6]. In most pelagic longline fisheries, discarded sharks are not reported in fisher logbooks [1], therefore their numbers, species composition and associated fishing mortality are poorly known.

Not all captured sharks are discarded. Several species are increasingly retained or have become secondary target species of pelagic longline fisheries [7]. Shark meat markets have shown an upwards trend over the past decades [8], and the demand for shark fins remains high [9]. Landings of some shark species have increased as a result of targeting [10], even though global shark and ray landings have declined by 20% since 2003, mainly as a result of fishing pressure [11]. Landings data alone grossly under-represent shark fishing mortality associated with pelagic longline fisheries, because it does not include discard mortalities. More accurate estimates of fishing mortality rates and levels, important for stock status assessments, can only be obtained by combining landings and discard mortality estimates [1, 2].

Advances in gear technology (satellite navigation; fishing vessel construction; polyamide monofilament line) have increased the selectivity of longline fishing gear, which can be adapted to target specific species groups [12]. Vessels can switch between fishery targets by setting hooks at different depths or times [5, 13], changing fishing locations [10], replacing gear components such as leaders on hooks [14] or by using different bait types [15, 16]. He et al. [17] segregated dissimilar types of fishing effort in a Hawaiian-based pelagic longline fishery, based on a cluster analysis of the species composition of landings. The approach distinguished between selective fishing for tunas (*Thunnus* spp.) or swordfish (*Xiphias gladius*), and mixed-species fishing that included pelagic sharks. In addition to varying gear and deployment characteristics, clear differences among vessel clusters were revealed when the composition of landings was matched with the spatial distribution of longline sets.

Targeting of sharks and discard practices vary substantially across fishing fleets, in response to market demand or regulatory measures [2]. James et al. [18] showed that shark species-specific economic value is a key determinant of whether a shark is retained and processed or discarded, and that country (a proxy for regulatory environment) was also important. Operationally, factors that affect the decision to retain or discard sharks are the abundance of other target species, whether sharks are damaged or too small to process, availability of freezer space, retention or trade bans, or output controls such as upper catch limits [2, 6, 18]. Mandatory release regulations operate in some longline fisheries [19], but in others there are landing obligations for species shown to have high potential for discard mortality [4].

Pelagic shark populations are typically vulnerable to overfishing because of life-history traits that include slow growth, low fecundity, late age at maturity, and a long natural lifespan [13, 20, 21]. Species such as blue sharks (*Prionace glauca*), shortfin makos (*Isurus oxyrinchus*) and porbeagle sharks (*Lamna nasus*) migrate freely and widely over their range [22, 23] making them vulnerable to both high-seas fishing fleets and local fleets that operate closer to the coast. The conservation status of several species is considered to be vulnerable with decreasing population trends (www.iucnredlist.org) and some species are listed on the Convention on International Trade in Endangered Species of Wild Fauna and Flora Appendix II (www.cites.org), which limits international trade.

In spite of the vulnerability of pelagic sharks to longline fishing, the quality and availability of reliable data on shark fishing mortality remain limited [2], compromising efforts to determine stock status. Campana et al. [1] highlighted the importance of independent observer programmes

to collect quantitative information on catch and discard rates by species, gear type and ocean region, to allow for more accurate estimates of fishing mortality for pelagic sharks.

The South African-flagged pelagic longline fishery lands tunas, swordfish and pelagic sharks caught in the coastal southeast Atlantic Ocean and southwest Indian Ocean. The bulk of blue sharks and shortfin makos landed by the fishery originate from the temperate south and west coasts of South Africa, with lesser quantities captured along the subtropical east coast [10, 24]. Sharks have been managed as a bycatch of the fishery since 2005 [19] and landings are reported to the International Commission for the Conservation of Atlantic Tunas (www.iccat.int) and the Indian Ocean Tuna Commission (www.iotc.org). Shortfin mako landings and catch per unit effort (CPUE) have increased in recent years, suggesting increased targeting and/or retention as part of a mixed-species fishing strategy [10]. Blue shark CPUE has varied, suggesting occasional targeting and retention, interspersed with periods when blue sharks were not caught or discarded and not reported.

A recent review of formal management protocols for sharks specified in the National Plan of Action for South Africa [25] recognized that management so far has been on an *ad hoc* basis [26]. The estimation of shark discards was highlighted as a weakness, because there were insufficient data to enable quantification of shark mortalities associated with bycatch. We addressed this limitation for blue sharks and shortfin makos captured by the South African-flagged pelagic longline fishery by estimating shark fishing mortality, as the sum of reported landings and discard mortalities estimated from observer data collected at sea.

## Materials and methods

### Study area

The South African exclusive economic zone (Fig 1) was stratified into four geographical areas: West (cool temperate waters influenced by the Benguela Current; Namibian border to 33˚S); Southwest (dynamic boundary zone between the Benguela Current and subtropical Agulhas Current systems, including the western Agulhas Bank; 33˚S– 20˚E); South (lower Agulhas Current area, where the narrow shelf broadens towards the west to form the eastern Agulhas Bank; 20–26˚E); and East (subtropical waters influenced by the upper Agulhas Current; 26˚E to Mozambique border). The four areas covered the main commercial fishing grounds used by the South African-flagged pelagic longline fleet (Fig 1) and conformed to a spatial framework used by Petersen et al. [24] and Jordaan et al. [10] to study shark bycatches of the fishery.

### Landings- and logbook data

Landings data on a per-trip basis were obtained from the Department of Environment, Forestry and Fisheries (DEFF) for South African-flagged pelagic longline vessels for 2013–2015. The data were comprised of the numbers and total weight of retained fish and sharks, categorized to species level, or to species groups for less common or similar-looking species. Blue sharks and shortfin makos dominated shark landings. Sharks that could not be identified to species level were grouped as requiem sharks (mostly *Carcharhinus* spp.), threshers (*Alopias* spp.), hammerheads (*Sphyrna* spp.) or as unidentified sharks.

We checked the consistency of landings data by dividing the total weight (processed weight) of landings per vessel by the numbers of fish or sharks reported, to obtain average individual weights. Average weights remained within plausible bounds for tunas (17–39 kg per fish) and swordfish (40–100 kg) with head, gut and fins removed, and for shortfin mako (6–37 kg) and blue shark (5–28 kg) trunks and fins (head and gut removed). Further, we regressed the recorded numbers per species group against the respective landed weights for individual vessels, expecting that landed weight would increase concurrently with increasing numbers. Least

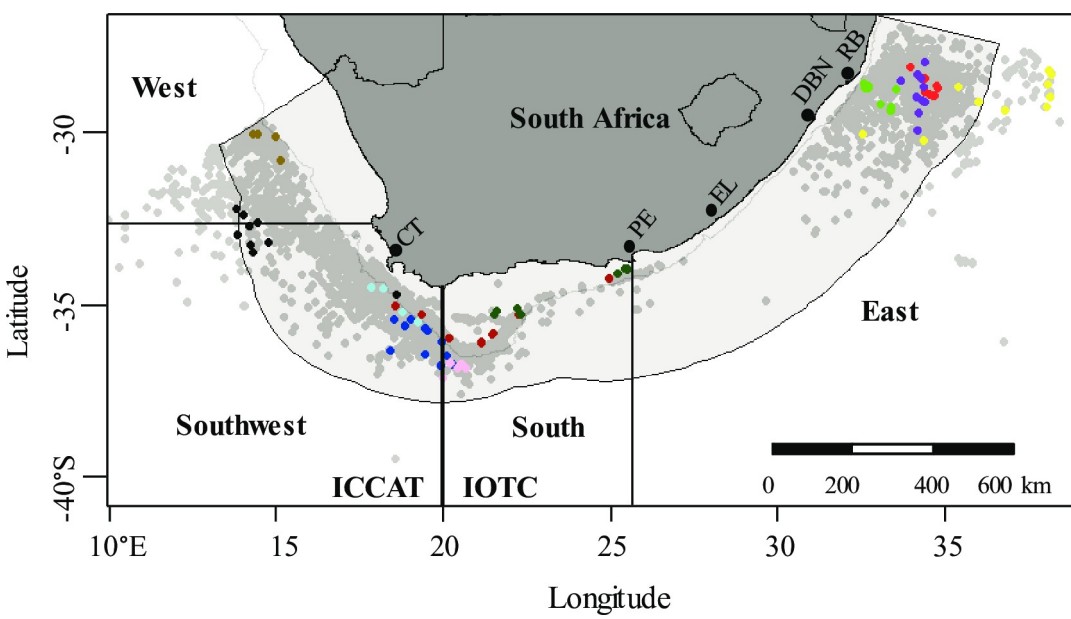

**Fig 1. Spatial distribution of sampled longlines per trip (coloured dots represent longline sets sampled by a fisheries-independent observer; n = 89 longlines) and of all longlines set in 2013–2015 (grey dots; n = 3 835).** Fishing areas were West and Southwest (SE Atlantic), South and East (SW Indian Ocean) [10, 24]. The boundary at 20˚E separated fishing areas reporting to the Indian Ocean Tuna Commission and the International Commission for the Conservation of Atlantic Tunas. The South African Exclusive Economic Zone extends 200 nm from the shore.

squares linear regressions confirmed strong positive relationships (n = 17 vessels; $r^2 > 0.900$; $p < 0.001$) for all four species groups tested. The tests confirmed that the landings data were of a consistent quality.

To obtain information on the spatial and temporal distribution of fishing effort, logbook records of individual fishing vessels (numbers of hooks set per day, set and haul positions) were extracted from the DEFF database. Retained catches per species group were also recorded in logbooks, but discarded individuals were not recorded. The dataset was cleaned by removing anomalous records in which fishing effort or catch composition were clearly incorrect or mismatched, as described in detail by Jordaan et al. [10].

## Fleet stratification using cluster analysis

We used a hierarchical cluster analysis and dendrograms in the statistical package R, version 3.3.2 [27] to stratify the pelagic longline fleet into subfleets, based on the relative proportions of tunas, swordfish, blue sharks and shortfin makos landed between 2013 and 2015. The 'hclust' function of the 'fastcluster' package in R [28] was used, relying on Euclidian distances between categories for each vessel and complete linkages [29]. We experimented with three and four *a priori* clusters in the dendrograms, and also explored hierarchical clustering among subsets of vessels landing mainly sharks.

A K-medoids clustering approach ('pam' function from the R-package 'cluster' [30]) was conducted independently from the hierarchical analysis and produced similar clusters (https://www.datanovia.com/en/lessons/k-medoids-in-r-algorithm-and-practical-examples/) [31] for the combined 2013 to 2015 data. An optimal fit of four clusters was derived from the average silhouette method ('fviz_cluster' function in R-package 'factoextra' [32]).

Individual cluster plots were generated for landings data in each year separately, to dynamically regroup vessels based on their annual landings. Movements of individual vessels between

subfleets in successive years (i.e. change of fishing behaviour based on landings information) could therefore be traced.

## Observer sampling

A fisheries-independent observer sampled the numbers of hooks hauled, and numbers of tunas, swordfish, and sharks caught during commercial fishing operations. All sampled vessels were in possession of a legal fishing permit, under the jurisdiction of DEFF. The observer underwent extensive training before embarking on vessels. No further ethics statements were required, because sampling was restricted to observation of fishing practices on commercial vessels, and no biological samples were collected.

Observer placement depended on opportunity and space availability on fishing vessels and was not randomly distributed. Nevertheless, different fishing vessels were sampled to facilitate broad coverage of discard practices and to ensure that all geographical areas and seasons were represented (Table 1). Observer sampling took place in 2015, with an additional sampling trip (4 sets) undertaken in 2018, to increase the sample size for subfleet 2. The vessel sampled in 2018 followed a fishing strategy typical to subfleet 2 (area fished; composition of landed catch), confirmed by the fishing company (species to be retained; area targeted; gear used), direct observation during the trip, and post-hoc comparison to data collected in 2015.

Sharks were identified to species level based on standard species identification guides [33]. Sharks were categorized as retained (kept and processed) or discarded (thrown overboard or branchlines cut adjacent to the vessel). Discarded sharks were further categorized as dead, alive in poor physical condition (debilitating injuries resulting from hooking in the eye or gills, clubbing, cutting or gaffing during handling, or motionless and unable to swim) or alive in good physical condition (shark clearly active, with no or minor physical injuries, such as a hook lodged in the jaw or mouth).

The observer data were used to calculate a catch ratio (number of sharks/number of sampled hooks) per vessel for retained sharks and for those in the three discard fate categories (i.e. discard ratios for sharks that were dead, alive in poor condition, or alive in good condition). To obtain catch ratios for unsampled vessels, data from sampled vessels were averaged, as follows:

$$\overline{y}_t = \sum_{n}^{n} y_i \tag{1}$$

$$s_i = \sqrt{\sum_{\frac{i=1}{n-1}}^{n} (y_i - \overline{y})^2} \tag{2}$$

**Table 1. Observer sampling effort per subfleet as number of sampling trips, quarter year covered, longlines and hooks sampled, and observed catches as numbers of tunas and swordfish, shortfin makos, blue sharks and other sharks.**

| Fleet | Sampling Effort | | | | | Observed Catch (numbers) | | | |
|---|---|---|---|---|---|---|---|---|---|
| | Trips (n) | Quarters covered | Longlines (n) | Hooks (n) | % hooks sampled | Tuna, Swordfish | Shortfin mako | Blue shark | Other sharks |
| Subfleet 1 | 7 | 1–4 | 63 | 45 925 | 56 | 1 048 | 100 | 1 703 | 101 |
| Subfleet 2 | 2 | 2, 4 | 8 | 6 524 | 61 | 13 | 19 | 1 281 | 2 |
| Subfleet 3 | 1 | 1 | 9 | 10 200 | 80 | 21 | 230 | 1 002 | 14 |
| Subfleet 4 | 1 | 1 | 9 | 8 454 | 71 | 0 | 1 185 | 258 | 124 |
| **All vessels** | **11** | | **89** | **71 103** | **61** | **1 082** | **1 534** | **4 244** | **241** |

Quarter years are January to March (1), April to June (2), July to September (3) and October to December (4).

Where $y_i$ is the discard ratio for vessel $i$, $\overline{y_t}$ is the average ratio and $s_i$ the standard deviation of catch per hook for each shark species and fate combination per subfleet. Catch ratios were only determined for 2015, because observer data were restricted to that year. Catch ratios were assumed to remain constant for extrapolations based on 2013 and 2014 fishing effort.

Catch ratios were used to scale observed numbers of individuals up to the total fishing effort based on logbooks, accounting for subfleet and discard fate category, as follows:

$$\hat{t}_{ST} = \sum_{i=1}^{L} N_i \overline{y_t} \tag{3}$$

$$\hat{V}\left(\hat{t}_{ST}\right) = \sum_{i=1}^{L} N^2{}_i \left(\frac{N_i - n_i}{N_i}\right) \frac{s^2{}_i}{n_i} \tag{4}$$

where $\hat{t}_{ST}$ is the stratified total numbers with variance, $\hat{V}\left(\hat{t}_{ST}\right)$ for each species and discard fate category [34], $N_i$ is the total number of hooks hauled per year, and $n_i$ are the numbers of hooks observed at sea. In the equation, fishing effort ($N_i$) could be reassigned between subfleets in each year between 2013 and 2015, to recreate the effects of vessel movements among subfleets. Reassignment took place on a per vessel basis (number of hooks set by individual vessel moved to appropriate subfleet categories in each year).

The accuracy of the estimation method was determined as the ratio of retained sharks ($R$, estimates obtained by raising observer counts to total fishing effort) to landed sharks ($L$, numbers of sharks landed, obtained from DEFF landings data). The ratio $R/L$ = 1.0 then indicates that retained sharks equal reported landings; R/L >1.0 indicates an overestimate, and R/L< 1.0 an underestimate.

## Results

### Cluster analysis and spatio-temporal distribution of fishing effort

The cluster analysis of landings data identified four subfleets based on the combined 2013 to 2015 data (n = 20 vessels): vessels that landed tunas and swordfish, but few sharks (subfleet 1; n = 6 vessels); vessels that landed tunas, swordfish and sharks (subfleet 2; n = 6); vessels that landed blue sharks and shortfin makos (subfleet 3; n = 4); and vessels that landed mainly shortfin makos (subfleet 4; n = 4) (Fig 2). Clustering of landings for each year individually identified the same four subfleets in all cases, with some vessels in each year exiting or entering the fishery, or moving between subfleet categories (S1–S3 Figs). The numbers of active vessels per year were 15 in 2013, 16 in 2014, and 17 in 2015. The cluster analysis therefore supported a distinct fishing behaviour per subfleet, based on the relative composition of landings.

A spatial analysis of fishing effort (numbers of hooks set) recorded in logbooks (n = 4.97 million hooks for 2013–2015 pooled data) showed that subfleets operated in distinct fishing areas. Subfleet 1 (tunas and swordfish) set >50% of hooks in the East, but no other subfleet fished in that area (Table 2). Subfleet 4 (shortfin makos) set most hooks in the South (58%) and Southwest (38%), whereas subfleet 3 (blue sharks and shortfin makos) set 77% of its hooks in the Southwest. Vessels that fished in the West (27% of hooks set by subfleet 1 and 52% set by subfleet 2) landed mainly tunas and swordfish, with a lesser proportion of sharks. Overall, tuna- and swordfish directed vessels (subfleets 1 and 2) fished mainly in the East and West fishing areas, whereas shark-directed vessels (subfleets 3 and 4) fished predominantly in the South and Southwest. There was considerable overlap between the spatial fishing pattern and landings, which could not be resolved at the coarse spatial scale used in the analysis.

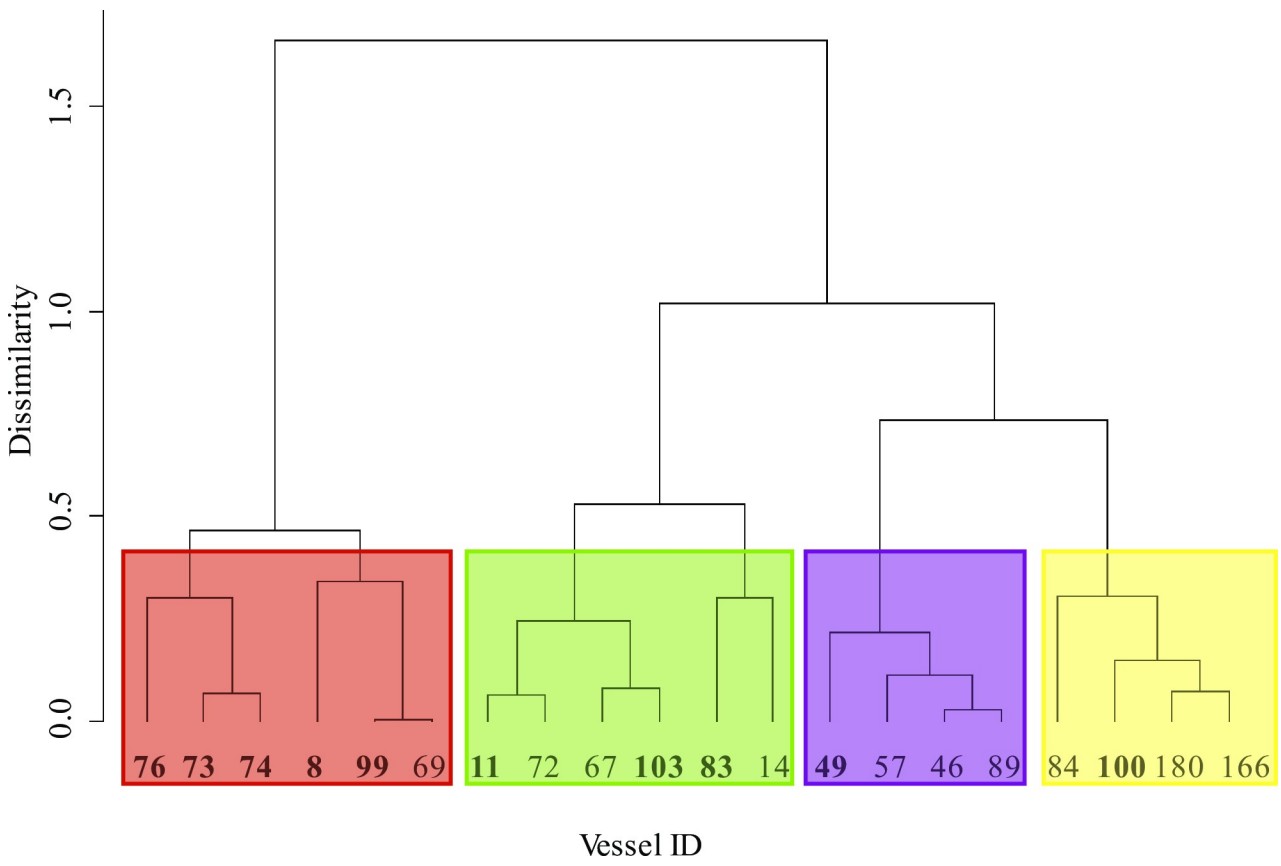

**Fig 2. Dendrogram of the vessels (n = 20) clustered into four subfleets based on the species composition of landings (2013–2015).** Vessels in bold were sampled by a fisheries-independent observer. Subfleet 1 (red) landed mainly tunas and swordfish; Subfleet 2 (green) landed tunas, swordfish and sharks; Subfleet 3 (purple) landed shortfin mako and blue sharks; Subfleet 4 (yellow) landed mainly shortfin makos.

Seasonal distribution of fishing effort, based on four three-monthly quarters in each year, differed among the four subfleets (Fig 3). Subfleets 1 and 2 set the fewest number of hooks in Jan-Mar (16% and 8% of annual effort, respectively) with the bulk of their fishing effort concentrated in Apr-Sep (59% and 68% of hooks set, respectively) to target tunas during winter months. Fishing effort of subfleet 3 (blue sharks and shortfin makos) was sharply down in Jul-Sep (13% of annual effort) but remained relatively consistent between 26% and 33% per quarter during the rest of the year. Fishing effort of subfleet 4 (shortfin makos) was consistent throughout the year, remaining within a narrow band of 23–28% of hooks set per quarter year.

## Observer samples

The observer sampled 71 102 of 116 872 hooks (61%) set along 89 longlines during 11 trips at sea (Table 1), on 10 different fishing vessels. All four subfleets were sampled although sampling

**Table 2. Spatial distribution of fishing effort per subfleet as percentage of hooks set per fishing area, based on logbook data from 2013–2015 (n = 4.97 million hooks).**

| Fleet | Composition of landings | West | Southwest | South | East |
|---|---|---|---|---|---|
| Subfleet 1 | Tuna & swordfish | 26–50 | 10–25 | <10 | 51–75 |
| Subfleet 2 | Tunas, swordfish & sharks | 51–75 | 26–50 | <10 | <10 |
| Subfleet 3 | Shortfin makos & blue sharks | 10–25 | 76–100 | <10 | <10 |
| Subfleet 4 | Shortfin makos | <10 | 26–50 | 51–75 | <10 |

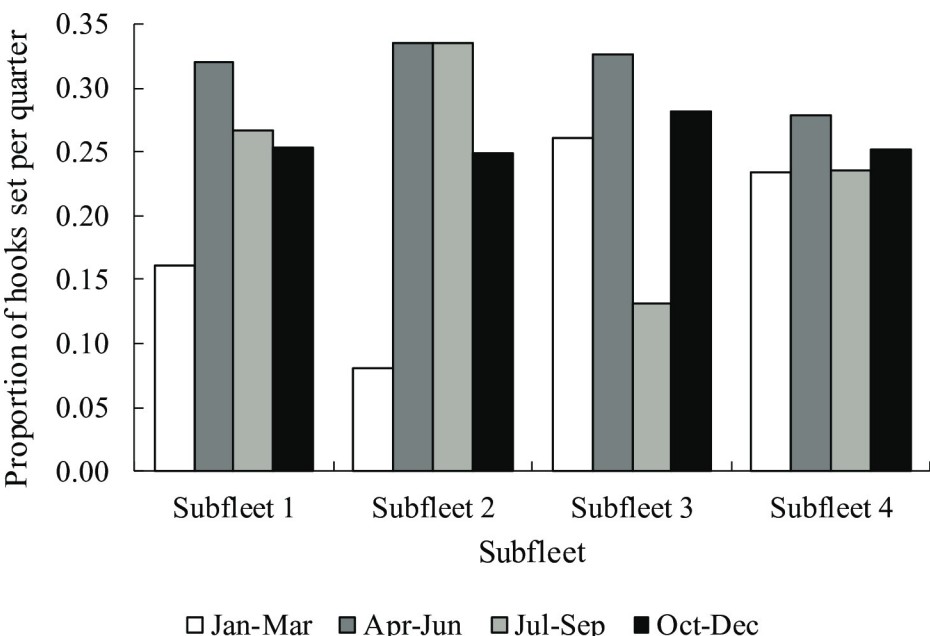

**Fig 3. Seasonal distribution of fishing effort by individual subfleets as the proportional numbers of hooks set per quarter year, based on 2013–2015 logbook data (n = 4.97 million hooks).**

intensity was unequal. The bulk of sampling was directed at subfleet 1 (65% of sampled hooks), where seven sampling trips were undertaken on six different vessels (one vessel was sampled twice). Two of four active vessels in subfleet 2 were sampled (9% of hooks); one of four in subfleet 3 (14% of hooks), and one of three in subfleet 4 (12% of hooks). By area, most hooks were sampled in the East (45% of hooks sampled), followed by 29% in the South, and 13% in both the Southwest and West areas. Observer coverage in the East was limited to the northern half of that fishing area, consistent with the spatial distribution of commercial fishing effort (Fig 1). By quarter year, four sampling trips were undertaken in January-March, two in April-June, one in July-September, three in October-December, and one trip started in June (end of quarter 2 and ended in July (beginning of quarter 3) (Table 1)). Overall, 5% of all commercial longlines set during 2015 (n = 1 699 lines set) were sampled.

The observed catches in the West and Southwest fishing areas were comprised mainly of blue sharks ($\geq$88% by numbers) and smaller quantities of shortfin makos ($\geq$2%), tunas (3%) and swordfish ($\geq$2%) (Fig 4). In contrast, shortfin makos dominated catches in the South (58%), but blue sharks remained relatively abundant (36%). In the East, mostly tunas (43%), swordfish (30%) and blue sharks (26%) were observed, with few shortfin makos. Blue sharks were common in all four areas, and made up the bulk of the observed catch in the pooled data (62% by numbers) followed by shortfin makos (22%). Numerically, tunas and swordfish combined made up only 16% of the observed catches.

A total of 6 019 sharks were sampled by the observer, comprising nine species and two species groups (hammerheads and thresher sharks) (Fig 5). Blue sharks (71%) and shortfin makos (25%) dominated shark catches, and of the remainder, bronze whaler *Carcharhinus brachyurus* (2%), silky sharks *C. falciformis* (1%) and thresher sharks *Alopias* spp. (1%) were present in smaller numbers. Other shark species were infrequently caught, making up <1% of all sharks observed.

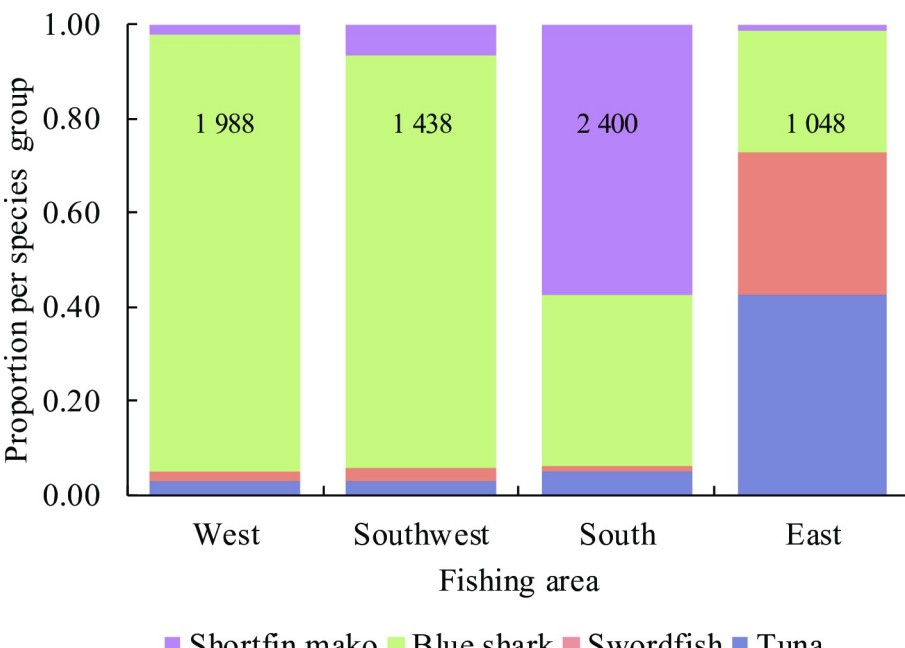

**Fig 4. Catch composition of key species groups per fishing area based on observer samples.** Sample size (number of sharks) per area is shown. Key species = Shortfin mako shark (*Isurus oxyrinchus*), Blue shark (*Prionace glauca*), Swordfish (*Xiphias gladius*), and Tuna (*Thunnus* spp.).

Of all hooked sharks, 47% were retained, 31% were discarded dead, 2% in poor physical condition, and 20% in good condition (Fig 5). Some 96% of shortfin makos and 32% of blue sharks were retained and processed, and the remainder discarded overboard. All other sharks were discarded, except for bronze whaler sharks, of which five individuals (5%) were retained by one vessel. Discard mortality rates differed among species (Fig 5). The bulk of discarded shortfin makos were dead (82%), and most discarded silky and thresher sharks were either dead (67% in both cases) or in poor condition (5%). Of 2 877 discarded blue sharks, 58% were dead and a further 4% in poor condition. Some 51% of bronze whalers, 54% of porbeagle, 32% of hammerheads and 25% of oceanic whitetip sharks were dead when discarded. Crocodile sharks were more hardy, and although none were dead when discarded, one individual (<10%) was in a poor condition. A single dusky and a tiger shark were captured and discarded in good condition.

## Estimation model

The ratios of retained sharks (estimated) over landed sharks (observed) (R/L ratios) were first compared between estimation models performed on unstratified data (single fishing fleet with homogenous fishing behaviour assumed) and stratified data (subfleets with heterogenous behaviour, based on outcome of cluster analysis) (Table 3). Stratification improved the accuracy of blue shark estimates, by increasing the R/L ratio from 0.46 to 0.86, relative to the benchmark DEFF landings data. For shortfin makos, the R/L ratio increased from 0.75 for the unstratified estimate to 1.27 for the estimate based on a stratified fleet. Stratification therefore increased the numbers of shortfin makos in estimates. The over-estimation of shortfin mako numbers after stratification (R/L = 1.27) suggests that the observer sampled trips with atypically high shortfin mako catches, especially in subfleets 3 (blue sharks and shortfin makos) and 4 (shortfin makos). The bias is therefore attributed to under-sampling (a single trip sampled in

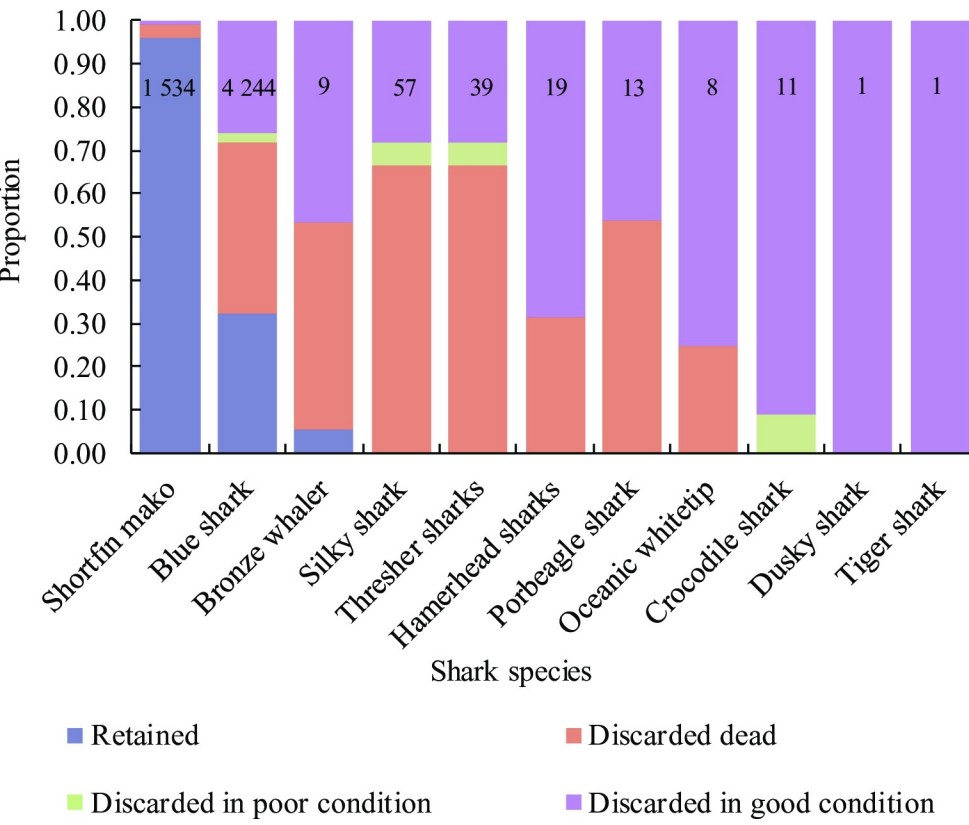

**Fig 5. Proportion of sharks per species or species group that were retained, discarded dead, in poor condition and in good condition, based on observer samples.** Sample size per species is shown.

subfleets 3 and 4, respectively; Table 1). Despite the bias, the relative accuracy of the estimates supported a hypothesis that catch ratios derived from observer data can successfully be combined with fishing effort data from commercial logbooks to reconstruct the numbers of sharks discarded without records.

Reconstructions of retained sharks by individual subfleets were more variable than for the pooled data. For subfleet 1, estimates were closest to benchmark landings data for blue sharks (R/L = 1.33) and shortfin makos (0.84) (Fig 6A and 6B), most likely because observer sampling was more extensive (7 trips on 6 vessels) and covered all four seasonal quarters. For this section, sharks discarded in poor physical condition (assumed unlikely to survive) included individuals that were discarded dead (see bars on Fig 6). The estimates confirmed that few shortfin makos were caught by subfleet 1 (est. 2 206 shortfin makos caught), and that only 17% of them were discarded, often dead or in a poor physical condition (est. 368 shortfin makos discarded, 49% in poor condition). Conversely, subfleet 1 caught blue sharks in large numbers (est. 35

**Table 3. Accuracy (R/L ratio) of reconstructed estimates of retained sharks relative to reported landings for unstratified versus stratified pelagic longline fleets.**

| Shark species | Landings in 2015 (n) | Unstratified est. (n) | R/L ratio | Stratified est. (n) | R/L ratio |
|---|---|---|---|---|---|
| Blue shark | 62 235 | 28 464 | 0.46 | 53 404 | 0.86 |
| Shortfin mako shark | 37 946 | 28 496 | 0.75 | 48 340 | 1.27 |

A ratio of estimated retained catch to observed landings (R/L ratio) of 1.0 indicates no difference, >1.0 indicates that estimated retained sharks exceed observed landings, and < 1.0 that it underestimates observed landings in 2015.

785) but discarded 96% of them. Of the discarded blue sharks, 35% were dead or in a poor physical condition. Even though subfleet 1 landed mainly tunas and swordfish, discarding of incidentally caught blue sharks, often dead or in a poor physical condition, indicated high fishing mortality of blue sharks.

For subfleet 2, the model underestimated retained catches of blue sharks (R/L = 0.54) and shortfin makos (0.35) relative to landings data (Fig 6C and 6D). Few shortfin makos were caught or discarded, but the numbers of blue sharks caught were exceptionally high (est. 69 687) of which 87% were discarded, nearly all of them dead or in a poor physical condition. Despite landing sharks together with tunas and swordfish, the high incidental catches of blue sharks that were discarded overboard indicated that subfleet 2 also had a high impact on blue sharks.

For subfleet 3, the model accurately estimated retained blue shark catches relative to landings data (R/L = 0.98) but underestimated retained catches of shortfin makos by about half (0.47) (Fig 6E and 6F). For subfleet 4, the retained blue shark estimate was also accurate (1.05), but shortfin makos were grossly overestimated relative to landings (2.42) (Fig 6G and 6H). Fewer observer samples from which to derive catch ratios, and sampling of too few vessels in subfleets 2 to 4 compared to subfleet 1, could explain the higher variability and reduced accuracy of estimates. Subfleets 3 and 4 landed mainly sharks, and consequently few captured shortfin makos (4% and 3%, respectively) or blue sharks (12% and 3%) were discarded. The greater proportion of shortfin makos than blue sharks landed by subfleet 4, compared to subfleet 3, was because more shortfin makos were caught by subfleet 4, rather than an increase in blue shark discards. A spatial effect is therefore implied, in which subfleet 4 operated in fishing areas where shortfin makos were more abundant than blue sharks, i.e. in the South area.

Estimates of the numbers of shortfin makos discarded by the fleet remained stable at low levels between 2013 and 2015, but blue shark discards were high in all three years and increased moderately to a peak of approximately 100 000 sharks in 2015 (Fig 7). The numbers of retained sharks increased for both species over the 3-year period. The 2013 and 2014 estimates should be viewed as indicative only, because they relied on constant catch ratios per subfleet based on 2015 observer samples (no observer samples available for 2013 and 2014).

## Discussion

Assumptions made during this study were only partially met in most cases, with implications for numerical estimates and the interpretation of results. For the fleet stratification, it was assumed that fishing behaviour differed among vessels, and that the composition of landings could be used to group vessels with similar behaviour. Van Helmond et al. [35] showed that individual vessels or subfleets operating within the same management system (same rules) adopted distinct fishing behaviours, which resulted in different outcomes, or landings composition. In our study, distinct subfleets or fishing behaviour could be inferred from differences in landings composition (see also [17]), were consistent over the 3-year study period, and matched preferred fishing areas in posterior analysis. We did not attempt to refine subfleets further, by including vessel characteristics, differences in gear configuration, or seasonal and marketing dynamics (see [36, 37]).

The assumption that observer samples accurately represented discard behavior of individual subfleets could only be partially met. All six vessels of subfleet 1 were sampled at least once, and one of them was sampled twice during 2015. Samples were spread nearly evenly across quarter years, to reduce seasonal bias in estimates (Table 1). For this subfleet, numerical estimates matched benchmark values well, implying that discard ratios could accurately be determined from samples. The other three subfleets were sampled less extensively, resulting in

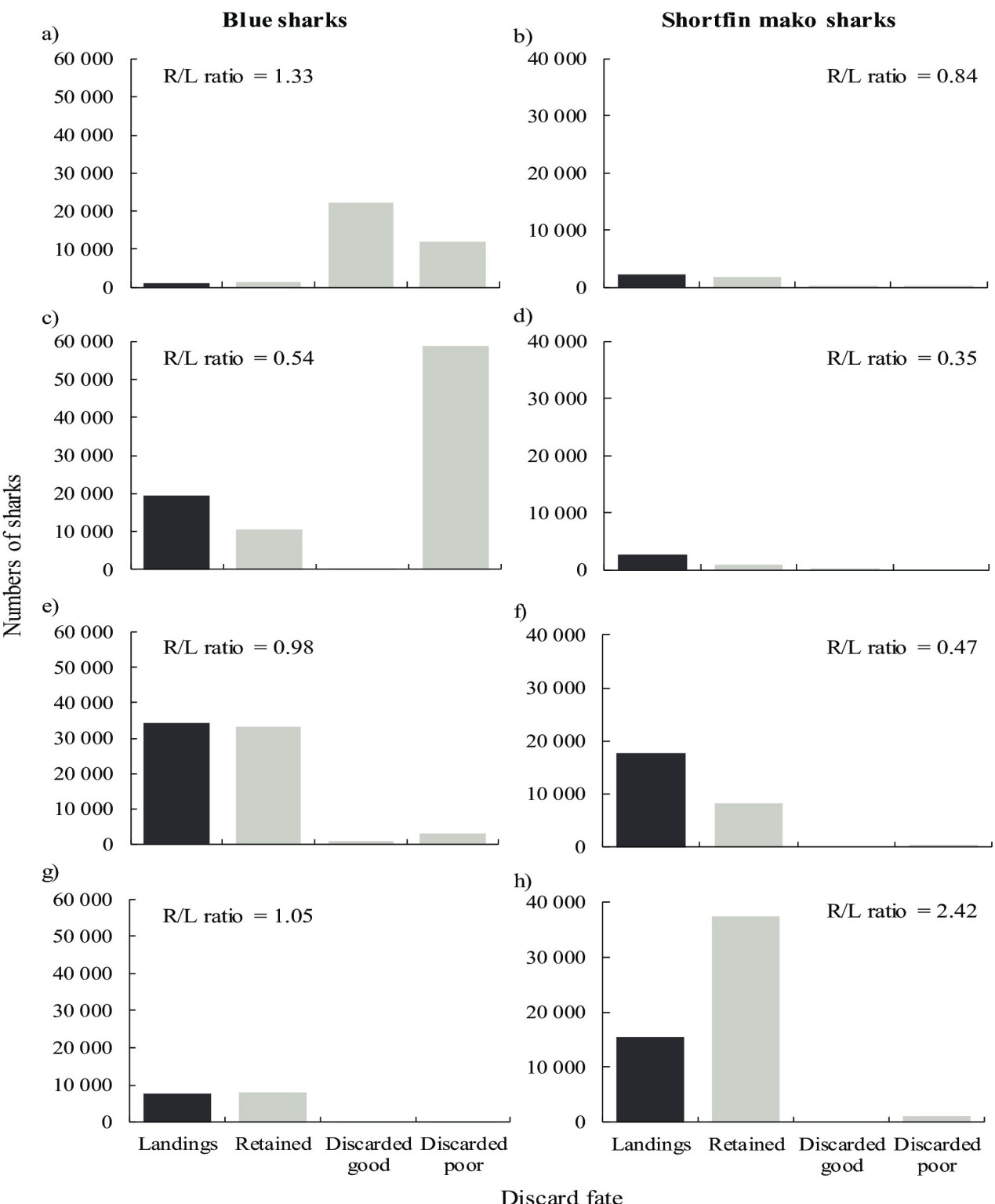

**Fig 6. Numbers of blue and shortfin mako sharks landed (dark bar; L) and estimates (light bars) of sharks retained (R), discarded in good condition and discarded in poor condition (assumed dead) for subfleet 1 (a and b), subfleet 2 (c and d), subfleet 3 (e and f) and subfleet 4 (g and h) based on observer samples, and raised to total fishing effort per subfleet.** The R/L ratio (estimated retained / reported landed) is provided as a measure of accuracy. A R/L ratio of 1.0 signifies equality; >1.0 is an overestimate and <1.0 is an underestimate.

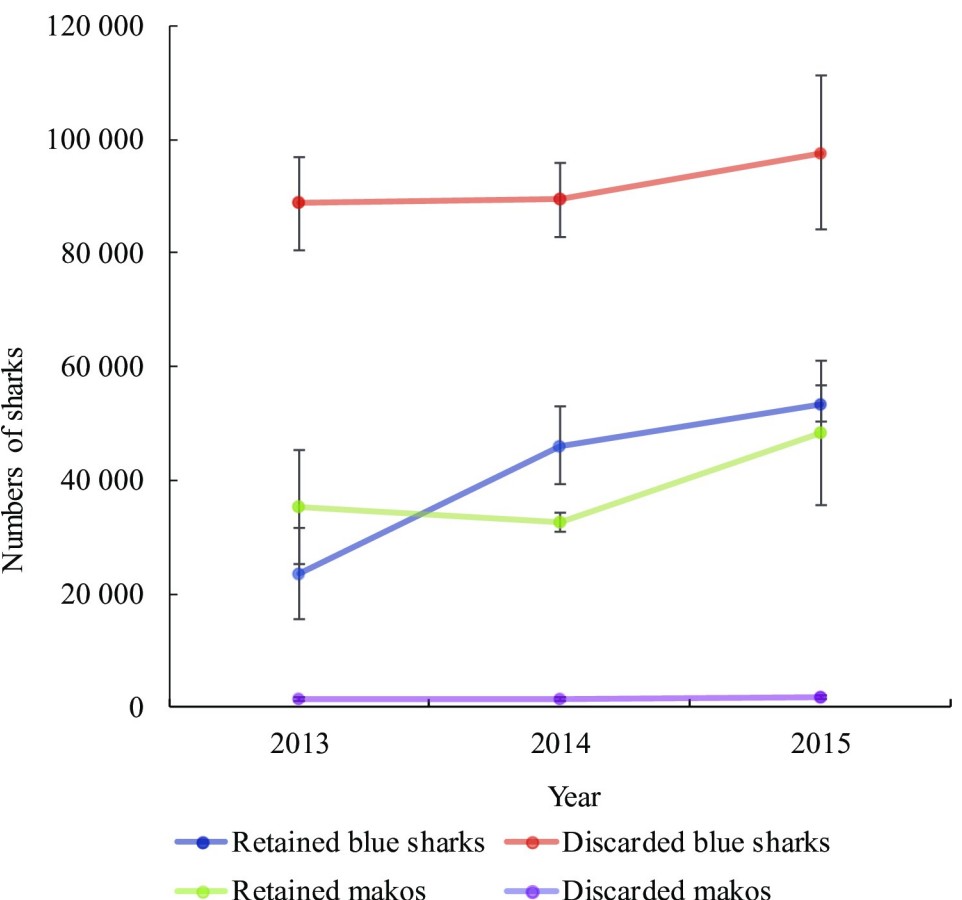

**Fig 7. Estimated numbers of retained and discarded blue sharks and shortfin makos (±S.E.) by the pelagic longline fleet in 2013, 2014 and 2015.**

much higher variability of discard ratios and an increased likelihood that atypical fishing behaviour during a sampling trip or seasonality would bias discard ratios. We therefore had less confidence in discard mortality estimates determined for subfleets 2 to 4.

We estimated that the South African-flagged pelagic longline fishery caught ~150 000 blue sharks and 50 000 shortfin makos during 2015. Of these, an estimated 24 000 blue sharks and <1 000 shortfin makos were released in a good physical condition. The combined fishing mortality (retained and discarded dead or in a poor condition) estimate of 175 000 sharks was nearly an order of magnitude greater than average mortalities of 16 000–22 000 sharks/year incurred by the same fleet in 1998–2005 [24]. The high fishing mortality estimates potentially include sharks subjectively categorized as in 'poor physical condition' (i.e. assumed dead) but which survived after release [38]. We therefore recognize that the high fishing mortality rates shown in this study may overestimate actual mortalities from this source.

An increase in fishing effort from 0.45 million hooks set in 2000 to 1.7 million hooks set in 2015 (nearly 4 times more; [10]) could partially explain the increase in shark fishing mortality. The present study and Petersen et al. [24] both extrapolated observer-based counts to total fishing effort, although fleet stratification and estimation methods differed. But more importantly, the two studies focussed on fundamentally dissimilar time periods regarding the importance of sharks on international markets, illustrated by a 10-fold increase in reported shark landings between 2003 and 2005 [10] to benefit from increased market prices during that

period [39, 40]. Despite differences in estimation methods used in the two studies, the steep increase in blue shark and shortfin mako fishing mortality over the past two decades is alarming.

Long-term trends in landings data and CPUE indices support the finding of a steep increase in shortfin mako fishing mortality [10]. Shortfin mako landings reported by the South African-flagged fleet increased from 869 sharks in 2000 to 37 946 in 2015, although the earlier landings may have been under-reported [24]. The increased landings originated mostly from expanded fishing grounds over the Agulhas Bank (South area) where shortfin makos were more abundant [41, 42]. Sharply rising CPUE indices after 2004 confirmed increased targeting of shortfin makos [10]. Our field samples showed that overall, only 4% of captured shortfin makos were discarded in 2015, hence landings data closely approximated fishing mortality, with discard mortalities contributing little.

Sporadic targeting of blue sharks, inferred from CPUE peaks and increased landings in some years [10], could partially explain the increase in blue shark fishing mortality. Nevertheless, the bulk of blue shark fishing mortality resulted from discards of captured sharks. Some 68% of captured blue sharks were discarded as unwanted catch during observer sampling in 2015, mostly by subfleets 1 and 2 (tuna and swordfish directed). Extrapolations from observer data suggested that of 36 000 blue sharks captured by subfleet 1, 96% were discarded, of which 35% were dead or in poor condition. Of 69 687 blue sharks captured by subfleet 2, 81% were discarded, nearly all dead or in poor condition (87%). The other two subfleets landed mainly sharks and therefore retained most blue sharks captured (est. 90% of 26 000 sharks caught by subfleet 3; 97% of 8 000 sharks by subfleet 4). Overall, selective fishing for tunas and swordfish contributed most to blue shark fishing mortality, because vessels discarded large numbers of sharks.

Most discarded blue sharks were either dead (58%) or in poor physical condition (4%). Our estimates were based on observations at discarding, and include mortalities resulting from the capture process (at-vessel-mortality or AVM) combined with those suffered during onboard handling, such as clubbing, cutting to remove hooks, long air-exposure times, or a combination of factors. Considering that published AVM values for blue sharks is generally <25% (reviewed by Ellis et al. [4]), on-board handling appears to have contributed substantially to discard mortalities in the present study. We do not rule out the possibility that blue sharks are purposefully killed before discarding on some vessels, to prevent them from depredating and damaging hooked fish after release [13]. The elevated blue shark mortalities may alternatively reflect fishing in nursery areas (G. Jordaan, pers. observation; [43]) where neonate or small juvenile blue sharks are presumably less likely to survive capture and handling than larger sharks.

Some 82% of shortfin makos in field samples were dead when discarded. The very high rate compared to the published AVM of 5–56% [4] was expected, because only badly damaged shortfin makos (depredated, decomposed) were discarded–the rest were kept (96% of catches) because of their comparatively high market value. Several other pelagic sharks were discarded by the fishery, because their retention onboard is legally prohibited [26]. Most silky sharks (67%), threshers (67%), porbeagle sharks (54%) and bronze whalers (48%) were dead when discarded. Our estimates moderately exceeded published AVMs of silky (56–66%), thresher (51–59%) and porbeagle sharks (21–44%) [4]. Oceanic whitetip and crocodile sharks were more hardy, and our estimates of 25% and 9%, respectively, compared well with published AVM ranges of 26–60% and 9–13% [4]. High variability in discard mortality rates is typical in pelagic longline fisheries, because time spent hooked, line configuration, hook type, handling practices, air-exposure time, and shark species, size or sex can all affect mortality rates [3–5, 13, 38].

Subfleet fishing strategies were strongly influenced by spatial considerations. Vessels fishing selectively for tunas and swordfish operated in the East, West and Southwest areas, and those landing primarily sharks frequented the South (shortfin makos) and Southwest (blue sharks and shortfin makos) (Table 2 and [10]). Studies from other ocean regions confirm that the spatial distribution of fishing effort determines the species composition of catches made by pelagic longlines [13, 44–47]. Hotspots of blue shark and shortfin mako abundance in the South and Southwest areas include blue shark nursery areas in the Benguela/Agulhas Current confluence [43, 44] and feeding grounds for juvenile shortfin makos near the Agulhas Bank edge in winter and spring [41]. The spatial and temporal dynamics of blue sharks and shortfin makos off South Africa, particularly in known nursery grounds, suggest that dynamic (event-triggered) spatio-temporal closures [48–50] can be considered as a fisheries management option to reduce blue and shortfin mako shark discard mortalities in vessels targeting sharks, tunas and swordfish.

Our study has several implications for the management of the South African-flagged pelagic longline fishery. Most pressing is the steep increase in fishing mortality of blue sharks and shortfin makos. Shortfin makos remained a primary target species of parts of the fleet, despite their official status as bycatch since 2005. A permit condition restricting shark landings to 60% of landed total mass in any quarter [51] is unlikely to succeed in changing fisher behaviour (see [37]), because this level of shark landings is set too high to effectively limit targeting of sharks. Shortfin makos have recently (2019) been listed on the Convention on International Trade in Endangered Species of Wild Fauna and Flora Appendix II [52] which will place stricter controls on international trade. The listing provides a clear incentive for reviewing shortfin mako mortalities and adjusting upper catch limits. Alternatively, shortfin makos can be managed as a primary fisheries resource, subject to sustainable management objectives. The prohibition of wire leaders on hooks [53] aims to reduce the incidence of sharks brought onboard, thus obviating the need to handle them. Zollett and Swimmer et al. [54] suggested that routine training of fishers on handling practices and the consequences of removing apex predators would reduce blue shark discard mortalities.

To conclude, stratification of the pelagic longline fleet based on landings composition (a proxy for fishing behaviour) provided a well-supported framework for observer-based sampling of discard ratios and estimation of unreported shark mortalities, especially for subfleet 1. Numerical estimates indicated a near 10-fold increase in shark fishing mortality compared to an earlier study (1998–2005), in agreement with upwards trends in fishing effort, reported landings and recent CPUE estimates [10]. Escalating shortfin mako fishing mortality was attributed to increased targeting to supply higher market demand. Discarding of blue sharks by selective fishing for tunas and swordfish had a greater impact on their fishing mortality than retention by shark-directed subfleets. Whereas the method developed for this study is rigorous, higher levels of observer sampling are required to increase confidence in discard ratio estimates.

## Supporting information

**S1 Table. The numbers of blue and shortfin mako sharks landed per subfleet in 2015 compared to the estimates of retained sharks, those discarded in good and poor condition, respectively including the standard error (S.E.).**
(DOCX)

**S1 Fig. Dendrogram of vessels clustered into four subfleets based on the species composition of landings in 2015.** Vessels shown in bold were sampled by a fisheries-independent observer. Subfleet 1 (red) landed mainly tunas and swordfish; Subfleet 2 (green) landed tunas,

swordfish and sharks; Subfleet 3 (purple) landed shortfin mako and blue sharks; Subfleet 4 (yellow) landed mainly shortfin makos.
(TIF)

**S2 Fig. Dendrogram of vessels clustered into four subfleets based on the species composition of landings in 2014.** Vessels shown in bold were sampled by a fisheries-independent observer in 2015. Subfleet 1 (red) landed mainly tunas and swordfish; Subfleet 2 (green) landed tunas, swordfish and sharks; Subfleet 3 (purple) landed shortfin mako and blue sharks; Subfleet 4 (yellow) landed mainly shortfin makos.
(TIF)

**S3 Fig. Dendrogram of vessels clustered into four subfleets based on the species composition of landings in 2013.** Vessels shown in bold were sampled by a fisheries-independent observer in 2015. Subfleet 1 (red) landed mainly tunas and swordfish; Subfleet 2 (green) landed tunas, swordfish and sharks; Subfleet 3 (purple) landed shortfin mako and blue sharks; Subfleet 4 (yellow) landed mainly shortfin makos.
(TIF)

**S1 Data. Landings data 2013–2015.**
(XLSX)

**S2 Data. Observer data.**
(XLSX)

## Acknowledgments

We thank CapFish, particularly Sihle Victor Ngcongo and Willem Louw, for their advice on how the pelagic longline fishing fleet operates and their efforts in organising the observer sampling trips for the first author (G.L. Jordaan). We thank the skippers and crew of the longline vessels for providing insights into various fishing strategies and techniques. Thanks are due to DEFF, particularly Charlene da Silva and Wendy West, for providing the landings and logbook data for the study, and for their advice. JS thanks the Faculty of Bio-Sciences, Fisheries and Economics and the Scientific Committee on Oceanic Research for their travel support.

## Author Contributions

**Conceptualization:** Gareth L. Jordaan, Jorge Santos, Johan C. Groeneveld.

**Data curation:** Gareth L. Jordaan.

**Formal analysis:** Gareth L. Jordaan, Jorge Santos.

**Funding acquisition:** Johan C. Groeneveld.

**Investigation:** Gareth L. Jordaan, Johan C. Groeneveld.

**Methodology:** Gareth L. Jordaan, Jorge Santos, Johan C. Groeneveld.

**Project administration:** Gareth L. Jordaan.

**Resources:** Gareth L. Jordaan, Jorge Santos, Johan C. Groeneveld.

**Software:** Gareth L. Jordaan, Jorge Santos.

**Supervision:** Jorge Santos, Johan C. Groeneveld.

**Validation:** Johan C. Groeneveld.

**Visualization:** Gareth L. Jordaan.

**Writing – original draft:** Gareth L. Jordaan, Jorge Santos, Johan C. Groeneveld.

**Writing – review & editing:** Gareth L. Jordaan, Jorge Santos, Johan C. Groeneveld.

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
