## [Decision Letter · Decision Letter 0]

9 Jun 2020

PONE-D-20-12467

Shark discards in selective and mixed-species pelagic longline fisheries

PLOS ONE

Dear Dr. Jordaan,

Thank you for submitting your manuscript to PLOS ONE. After careful consideration, we feel that it has merit but does not fully meet PLOS ONE’s publication criteria as it currently stands. Therefore, we invite you to submit a revised version of the manuscript that addresses the points raised during the review process.

Overall, I found this to be an interesting manuscript. However, I think the text does need some work. In addition, Reviewer 1 had some concerns about the data set used, and has suggested expanding this if possible. Reviewer 2 has provided relatively minor comments to improve the clarity of the text. I have also provided detailed editorial comments. The authors should consider and respond to all the comments provided when making their revisions.

We look forward to receiving your revised manuscript.

Kind regards,

Heather M. Patterson, Ph.D.

Academic Editor

PLOS ONE

Journal Requirements:

Reviewers' comments:

Reviewer's Responses to Questions

**Comments to the Author**

1. Is the manuscript technically sound, and do the data support the conclusions?

Reviewer #1: No

Reviewer #2: Yes

2. Has the statistical analysis been performed appropriately and rigorously? 

Reviewer #1: No

Reviewer #2: Yes

3. Have the authors made all data underlying the findings in their manuscript fully available?

Reviewer #1: Yes

Reviewer #2: Yes

4. Is the manuscript presented in an intelligible fashion and written in standard English?

Reviewer #1: Yes

Reviewer #2: Yes

5. Review Comments to the Author

Reviewer #1: There is significant concern over the lack of data from which the authors conducted their analyses. The time series is short, at only 3 years. It is evident from a previous publication from the same authors (Reference 10) that they have access to a 15 year time series. The reason for the exclusion of the additional 12 years is cause for concern.

Additionally, only one year of observer data is available, which is partial and not comprehensive of all seasons, fishing areas or vessels. Of the observed trips, only 56-80% of hooks were sampled. In three of the four subfleets, only 1 or 2 trips were covered, and therefore cannot be assumed representative of all similar trips. While a coverage rate of 5-6% for a single year is generally accepted, the lack of randomization, inclusion of all areas in all seasons and incomplete sampling effort reduces the reliability of the data substantially. Of all data collected, only Subfleet 1 could be considered robust enough for the amount of extrapolation conducted, and that is excluding the consideration that of the observed trips, only 56% of hooks were sampled.

There are multiple instances of subjective terms ("often", "nearly all", etc.) being used to categorize the data presented without supporting evidence or presenting values.

While the authors do include acknowledgement of lack of data, bias and assumptions not being met, it is unclear why all analyses were conducted and/or data included. The recent listing of shortfin mako sharks under CITES Appendix II may play a role in this publication. It is unclear if additional observer data is available, and if so why it was not included, in addition to the exclusion of the larger time series.

It is the hope of the reviewer that the authors will consider expanding the time series and include all data available for analysis instead of what appears to be picking and choosing information for presenting a predetermined conclusion.

Please see attached document for all comments.

Reviewer #2: I have reviewed the manuscript “Shark discards in selective and mixed-species pelagic longline fisheries” (PONE-D-20-12467) and am of the opinion that it is acceptable for publication after minor revision. Most of my comments are minor and are easily addressed (see below). I feel the manuscript provides information that will be useful for stock assessment purposes and will be of interest to a wide range of readers.

Line 24: It is not correct to say the conservation status of sharks is vulnerable as there are many species of sharks that are not impacted by anthropogenic activities. The term “sharks” is too broad in this context.

Should the scientific names of swordfish and the shark species mentioned within the abstract be stated at first use?

Line 25: There is quite a bit of fishing morality information in some regions. This statement is too broad and needs to be revised for accuracy.

Line 27: Should Xiphias gladius be inserted here?

Line 35: Insert “was” before “comprised”.

Line 44: Is the term “subfleets” needed? Does “fleets” not suffice?

Line 53: See comment for Line 25.

Line 56: Similarly, this statement is true for some areas but not necessarily others.

Line 62: Shouldn’t at vessel mortality be included as well?

Line 63: Would be informative to include a few examples of gear advancements parenthetically.

Line 72: So He only found differences related to set location? What about for sets conducted in the same area but with varying gear and deployment characteristics?

Line 74: Perhaps more clear to simply write “species-specific economic value”.

Line 83: Simply state they have a relatively late age at maturity and low fecundity rather than “few offspring that mature late”. The later can be interpreted in a number of ways.

Lines 85-88: This is overly vague and it would be more informative to provide examples.

Line 96: The words southeast and southwest should not be capitalized. Insert Ocean after Atlantic.

Line 129: This is not clear. It might be clearer to state the colored dots represent trips or sets where data were collected by observers rather than “individual trips”.

Line 137: Insert “were” after “data”.

Line 146: By discharged are the authors referring to landed weights?

Lines 148-149: Be consistent in use of significant digits.

Line 154: As sharks are fish I suggest replacing this with “individuals”.

Line 203: Same.

Line 206: Both italicized “I”s should be in subscript here and throughout.

Line 208: Change “between” to “among”.

Line 209: Should this be “per vessel basis”.

Line 231: Should this be sampled by “an” observer” rather than “the” observer”? Same for Line 234.

Table 2 needs further detail in the legend. For example, what do shaded areas represent? Does the range within each cell represent the range among years?

Line 253: Edit to “…. the fewest number of hooks…..”

Lines 257-258: Be clear how the quarters were defined. It’s clear when looking at Fig 3 but should be stated in the text.

Line 265: What is meant by lines? I assume this is the same as sets. I would use the latter throughout for consistency.

Line 273: Is this correct that 13% of hooks retrieved were sampled by observers in both areas? If so, I recommend not using the word “respectively” here. Clearer to write “…and 13% in both the ….”

Line 278: Delete “some”.

Line 280: Insert “was” before “comprised”.

Line 281: Including the catch composition for each species parenthetically, as was done for blue sharks, would be informative. I suggest doing the same throughout this section for each species or group mentioned.

Line 288: The scientific name of each species not mentioned within the text should be included within the legend.

Line 304: Include scientific name of each species at first mention. Many missing in this section. Also, including sample size parenthetically along with the associated percentages of each species within the section would be informative.

Line 305: Change “between” to “among”.

Line 307: Were 5% of both species in poor condition? If so, no need for “respectively”.

Line 314: Be clear what is being estimated.

Line 329: I think the legend needs further detail. For example, be explicit…., equality of what? Overestimate of what?

General: If available, differences in gear configurations associated with each fleet would be informative.

Line 366: I suggest using the name shortfin mako throughout the manuscript rather than using mako and shortfin mako interchangeably.

Line 389: Should this be a posteriori?

Lines389-391: Did the observers not record gear characteristics? Further explanation is needed about why these factors were excluded from analyses.

Line 402: Delete “some” and indicate that these are estimated values.

Line 404: There is literature describing that sharks subjectively assigned a poor release condition often survive (e.g. Sulikowski et al.). Some discussion about the possibility and the reasons for the possibility of overestimating fishing mortality is warranted.

Line 411: Change on to in.

I feel it is very important to discuss the differences in fishing practices among the subfleets if that is possible. This would provide important information and directions for future research to mitigate bycatch.

Line 477: The meaning of set too high is not clear. Please edit for clarity.

Line 483: Include those rates for comparison here.

6. PLOS authors have the option to publish the peer review history of their article (what does this mean?). If published, this will include your full peer review and any attached files.

Reviewer #1: No

Reviewer #2: No

---

## [Author Response · Author response to Decision Letter 0]

2 Jul 2020

REVIEWER 1

OVERALL COMMENTS

Reviewer 1: There is significant concern over the lack of data from which the authors conducted their analyses. The time series is short, at only 3 years. It is evident from a previous publication from the same authors (Reference 10) that they have access to a 15 year time series. The reason for the exclusion of the additional 12 years is cause for concern.

Authors: The study is based on two datasets: observer data of shark catches and discards collected mainly by the first author while at sea on commercial fishing vessels; and official landings data (including logbook data) obtained from the fisheries department. The observer data are the crux of the study; data were collected during 11 trips at sea undertaken in 2015 (some of them up to 2 weeks long) during which 89 longline sets were sampled (71 000 hooks) and 6019 sharks were observed (all of this summarized in the text, and clearly showed in Table 1). The second dataset (landings and logbook data) were used to stratify the fleet into subfleets, and to assess spatio-temporal distribution of fishing effort – during the period for which we had the observer data. 

It would have made no sense to collect observer data in 2015, but then rely on landings and logbook data collected 15 years ago, because targeting (i.e. subfleets) and the spatial distribution of fishing effort have changed enormously over this period (see Jordaan et al. 2018 – Reference 10). 

We did use landings and logbook data for the 2013 – 2015 period to test whether subfleets operated in a consistent manner over the period of our study, and to investigate the effects of movements of individual vessel between subfleets. Note that cluster analyses were conducted for each year individually (2013, 2014 and 2015 – see Figures in supporting information) and that data were only combined over the 3 years when it was very clear that subfleets followed a consistent fishing strategy over the period.

Figure 7 back-extrapolated the 2015 observer data to 2013 and 2014 (for which subfleets operated in a consistent manner – see explanation above) and raised estimated numbers of sharks retained and discarded accordingly. Back-extrapolating any further into the past would assume that fishing strategy remained constant over time (pre-2013), and that the 2015 observer samples accurately represented discarding behaviour in the past. These assumptions were tested and rejected by Jordaan et al. 2018 (Reference 10). The reviewer’s concern, that we cherry-picked the data, is therefore fully rebutted. 

Given the large amount of observer data collected by the first author (6019 sharks observed; 71 000 hooks observed), combined with official landings and logbook data for the entire fleet of 20 vessels for the 2013-2015 period (> 4 million hooks set), we contest the reviewer’s opinion that there was a “lack of data”. 

Reviewer 1: Additionally, only one year of observer data is available, which is partial and not comprehensive of all seasons, fishing areas or vessels. Of the observed trips, only 56-80% of hooks were sampled. In three of the four subfleets, only 1 or 2 trips were covered, and therefore cannot be assumed representative of all similar trips. While a coverage rate of 5-6% for a single year is generally accepted, the lack of randomization, inclusion of all areas in all seasons and incomplete sampling effort reduces the reliability of the data substantially. Of all data collected, only Subfleet 1 could be considered robust enough for the amount of extrapolation conducted, and that is excluding the consideration that of the observed trips, only 56% of hooks were sampled.

Authors: One year of observer data was collected by the first author (see explanation above and Table 1 in the manuscript). The comment that “only 56 – 80% of hooks were sampled” during observed trips does not take into account the practicalities of sampling commercial fishing at sea (a 24-hour, 7 days a week operation). Indeed, 56 – 80% of hooks sampled is an exceptionally high proportion of hooks sampled by any subsampling standards.

All quarters were sampled for Subfleet 1, and we agree fully with the reviewer that Subfleet 1 is the only one for which observer data were robust enough for the extrapolations that were conducted. The assumption that observer samples accurately represented discard behaviour (i.e. data robustness) is diligently unpacked in both the results section and in the discussion of the manuscript. Data shortcomings and their effects on estimates form the central discussion points of this study, as was made clear from the very first sentence of the Abstract, i.e.: “ The conservation status of several pelagic shark species is considered vulnerable with declining populations, yet data on shark fishing mortality remain limited for large ocean regions.” (Revised version). The last sentence in the Abstract similarly concludes that: “Higher levels of observer sampling are required to increase confidence in discard ratio estimates” 

We believe that the reviewer’s comment is fully addressed within the manuscript as it stands, at multiple levels. 

Reviewer 1: There are multiple instances of subjective terms ("often", "nearly all", etc.) being used to categorize the data presented without supporting evidence or presenting values.

Authors: Unfortunately the reviewer was not specific, thus making it difficult to justify each occurrence of ‘often’ or ‘typically’ or ‘nearly all’ in the manuscript. 

We do not agree that the use of ‘often’ is not justified in at least some of the cases, for example, in line 49 : “Hooked sharks often die during capture or shortly thereafter as a result of physical injuries….” Here it is exactly what we intended to say (often = frequently; many times; many cases). 

We do not understand (or agree) with the reviewer’s comment (in the margin of the MS) that using often for something that occurs 35% of the time is misleading. We could not find a percentage mentioned anywhere in a search for the word “often” in online dictionaries. 

Reviewer 1: While the authors do include acknowledgement of lack of data, bias and assumptions not being met, it is unclear why all analyses were conducted and/or data included. The recent listing of shortfin mako sharks under CITES Appendix II may play a role in this publication. It is unclear if additional observer data is available, and if so why it was not included, in addition to the exclusion of the larger time series.

Authors: The reviewer raises several points here. Firstly, we contend that “..the authors do include acknowledgement of lack of data etc.” is a very narrow interpretation of the study, seeing that the study actually addresses the lack of data and assumptions fully, from the first line of the abstract, right through to the end of the discussion.

Secondly, we do not understand the statement of the reviewer that it is “unclear why analyses were conducted and/or data included”. Data used and analyses done are extensively explained in the Materials and Methods section. 

Thirdly, the recent listing of shortfin makos happened long after the onset of the study in 2015 – it had no influence on the decision to publish.

And fourthly, our observer data were collected according to a set protocol, to allow for this specific study. Observer data collected during other studies focussed on other objectives, and were not compatible with our objectives. Hence it was not included in this study. 

Reviewer 1: It is the hope of the reviewer that the authors will consider expanding the time series and include all data available for analysis instead of what appears to be picking and choosing information for presenting a predetermined conclusion.

Authors: Expanding the time series is not possible because there is no long time-series of observer data, as explained above. We fully disagree with the reviewer’s insinuation that we ‘picked and chose information for presenting a predetermined conclusion’. The study was planned in advance with specific objectives and data protocols. The extensive text in the discussion explaining the potential influences of data limitations and assumptions made debunks the reviewer’s opinion, in this instance. 

LINE BY LINE COMMENTS (from marginal comments in pdf)

Line 25: Sentence changed

Line 39: No change required – comment only

Line 49: Use of “often”. See general comments

Line 51: The reviewer is referred to Reference 3 – 6 for multiple examples. Too much detail for an Introduction. 

Line 89-90: We removed the word “extremely” from the sentence and refer the reviewer to Reference 2, which based its findings on multiple tuna fisheries and an extensive literature review of discard rates in each fishery. 

Line 128-129: Agreed. Figure title has been rephrased to make it clearer.

Line 145: Agreed. The weights do appear low when compared to live animals, because they are processed weights (i.e. gut removed). It is very seldom (if ever) that catch (caught for export) is landed whole without being gutted, therefore the recorded weight is not the same as live weight. We have added “(processed weight after removing the gut)”to the sentence to clarify. 

Observer sampling: The reviewer’s concerns are addressed in the general comments

Line 177-179: The reviewer’s concerns over sampling strategy are addressed in the general comments. Greater 

variability in discard ratios for subfleets sampled less extensively are discussed in depth in the Discussion section. 

Line 220: Twenty individual vessels landed catch between 2013 and 2015. Reference 10 (61 vessels) refers to a much longer period (2000 – 2015) which is not relevant to the present study. 

Line 225: We have added ‘in 2015’ to the figure caption to indicate that the vessels in bold were sampled by the observers in 2015. Observers did not sample vessels in 2013 and 2014; only in 2015. See general comments. 

Line 226: Thank you, our mistake. Changed to 15 in 2013 and 16 in 2014.

Line 234-235: Agreed. Sentence deleted. Vessel 83 was given as an example of a vessel that moved between subfleets in our analysis, by adopting a different fishing strategy. But in hindsight it is not needed here in the text. 

Line 237-238: The use of data and the assumptions made are explained in detail in the general comments part above. 

Line 320-322: Results from subfleets 3 and 4 remain important within the context of the study, despite the lower sampling coverage. For example, it highlights that a higher observer sampling effort (as seen in subfleet 1) is required when estimating discards. Furthermore, even though there may be over-estimation, it highlights the high incidence of 

unreported discards from this fishery.

Lines 324-326: No comment. 

Line 353: The reviewer does not appreciate the realities of sampling at sea on a commercial longliner. Detailed explanation given in general comments above. 

Line 356: See Figure 6c. 

Lines 362-364: See comments made for lines 320-322.

Lines 368-369: A spatial effect is shown for specifically what this vessel landed, shortfin makos. The vessel concentrated its fishing effort in areas of high shortfin mako abundance. We elaborated in the discussion with relevant references.

Line 381: Explained in detail under general comments above

Line 404-406: All issues to do with data, bias and confidence have been rebutted in the general comments section and individual responses above. The bias and confidence inherent in the sampling are discussed in detail in the manuscript, highlighting the potential effects on estimates – i.e. we are confident in the estimates for subfleet 1, and less confident for subfleets 3 – 4, with limitations explained. Comparing results with other studies forms part of the interpretive process in science – without it any results produced by a study would have to be seen in isolation, and no progress would be made.

Line 431: Added in percentage.

Lines 441-442: This was witnessed by the observer (the first author GJ); and the shark’s condition when discarded was recorded. The study was not looking at the AVM therefore there was no need to record the state of the animal when brought on board.

Line 473-474: “Steep increase” is a valid statement despite data limitations for reasons mentioned above in line 381 and lines 404-406.

Line 483-484: See comments for lines 441-443.

Lines 489-491: Added in ‘especially for subfleet 1’.

Line 492: Change made. This was the authors mistake.

REVIEWER 2

Lines 24 and 25: Agreed. Sentence changed, please see revised manuscript with track changes.

Line 27: Scientific name added for swordfish, genus name added for tuna.

Line 35: Done.

Line 44: Changed to fleet. 

Line 53: Agreed. Changed the sentence as follows: ‘In most pelagic longline fisheries, discarded sharks are not reported in fisher logbooks [1], and their numbers, species composition and associated fishing mortality are therefore poorly known.’

Line 56: References 8 and 9 qualify the statement

Line 62: At vessel mortality rates are already included – within the context of the study they are either part of ‘discarded dead’ or as observed landings. No change made. 

Line 63: Agreed, examples added.

Line 72: Agreed. We checked the reference and then changed the sentence as follows: ‘In addition to varying gear and deployment characteristics, clear differences among vessel clusters were revealed when the composition of landings was matched with the spatial distribution of longline sets.’

Line 74: Done

Line 83: Done

Lines 85-88: Agreed. We now start the sentence as follows: ‘Species such as blue sharks (Prionace glauca), shortfin makos (Isurus oxyrinchus) and porbeagle sharks (Lamna nasus) migrate freely….’ 

Line 96: Suggestion inserted.

Line 129: Inserted ‘observer sampling’

Line 137: Added ‘were’ before comprised

Line 146: Replaced ‘’discharged weights’ with landed weights.

Line 148-149: OK, done

Line 154: Changed ‘fish and sharks’ to ‘individuals’.

Line 203: Changed ‘fish and sharks’ to ‘individuals’.

Line 206: Noted and changed.

Line 208: Noted and changed.

Line 209: Agreed. Changed to per vessel.

Line 231: No change made – most vessels were sampled by the first author – therefore ‘The observer’ (singular) was retained. 

Table 2: No change made. The range in each block (percentage of hooks set from 2013 – 2015) is already provided in the Table, and the graduated grey-scale is only used to guide the reader’s eye. We can remove the grey altogether, if needed, but prefer to keep it. 

Line 253: Agreed. Done

Line 257-258: Agreed. Added the following: ‘…based on four three-monthly quarters in each year….’ 

Line 265: Lines refer to sets. Changed this as suggested.

Line 273: Agreed. Changed.

Line 278: Agreed. Done.

Line 280: Agreed. Done. Note that we added ‘were’ not ‘was’ because it is plural. 

Line 281: Agreed. Done.

Line 288: Agreed. Scientific names added in legend.

Line 304: Scientific names already mentioned in lines 293-294.

Line 305: Noted and changed.

Line 307: Removed ‘respectively’.

Line 314: Agreed. We added ‘The ratios of retained sharks (estimated) over landed sharks (observed) (R/L ratios)….’ as a first sentence. 

Line 329: Agreed. Both the caption and the legend of Table 3 were improved 

General: No change made. Gear characteristics were variable between vessels and trips and changes were not always captured on data sheets. Therefore, we focussed on spatio-temporal aspects fishing effort and the relative composition of landings to explain fishing behaviour. Adding gear configuration to the study would have added another layer of complexity and uncertainty, which we doubt that the data available would have been able to support. 

Line 366: Done. Shortfin mako now used throughout the text.

Line 389: No change made. Posterior analysis and ‘a posteriori’ are the same thing – in English and Latin. We prefer the simplest route here.

Line 389-391: See comment under General above. 

Line 402: Done. Replaced ‘some’ with ‘an estimated’.

Line 404: Agreed. Added the following text and a reference for Sulikowski et al 2020: ‘The high fishing mortality estimates potentially include sharks subjectively categorized as in a ‘poor physical condition’ (i.e. assumed dead) but which survived after release (Sulikowski et al. 2020). We therefore recognize that the high fishing mortality rates shown in this study may overestimate actual mortalities from this source.’ 

Lines 411: Unclear about this. ‘In’ or ‘on’ international markets? We think it should remain ‘on’.

Differences in fishing practices among subfleets form the backbone of this study, and we have analysed spatial and temporal trends in fishing effort, the species composition of landings, discard practices at sea (i.e. which species retained and which discarded), relative CPUE and discard mortality rates by species. We are unclear as to which other fishing practices the reviewer refers to? 

Gear configuration was variable, depending on sea conditions and target species, and as stated above, the data available does not support a further analysis of gear configurations / other fishing practices beyond what we have already analysed at this point. Basically, we have now identified subfleets based on when and where they fish, and what they land. A next step (outside the scope of this study) can now be to compare differences in gear and vessel configuration, and hook sizes, bait, time of setting and hauling etc. We argue that it is too much for the present study. 

Line 477: Agreed. We edited the sentence as follows: “…because this level of shark landings is set too high to effectively limit targeting of sharks”

Line 483: We deleted this section (2 sentences) because it repeated the discussion in a paragraph above (lines 436 – 447 in the original manuscript). Percentages are given in that paragraph, as the reviewer requested. 

EDITOR

We have included an ethics statement and details on any permits and permissions that may have been needed for this study.

LINE BY LINE COMMENTS

Line 24: Sentence changed, please see revised manuscript with track changes.

Lines 30-31: Scientific names added.

Line 35: Noted and changed.

Line 60: Noted and changed.

Line 69: Scientific names added.

Line 82: Noted and changed.

Line 85: Noted and changed.

Line 87: Noted and changed.

Line 92: Deleted hyphen.

Line 96: I have removed the abbreviations. This is referring to the southwest Indian Ocean. Later on in line 122 Southwest refers to one of the geographical areas.

Lines 101 and 102: Removed abbreviations.

Line 102: Removed ‘respectively’.

Line 107: Changed to ‘in the National Plan of Action for South Africa’.

Line 108: Deleted NPOA.

Line 119: No this is a geographical region. Previously it was the southwest Indian Ocean.

Line 131: Spelled out IOTC and ICCAT

Line 132: Removed EEZ.

Line 137: Noted and changed.

Line 197: Removed ‘respectively’.

Line 219: Changed ‘recovered’ to ‘identified’.

Line 224: Changed ‘recovered’ to ‘identified’.

Line 225: Removed ‘Supplemental data’.

Lines 234-235: This is mentioned here because if one was to look at the raw data provided, they will see that the vessel was sampled in a different year. It is better to disclose this information here than not at all.

Line 246: Noted and changed.

Line 254: Added in ‘respectively’.

Line 256: Added in comma.

Line 273: Removed ‘respectively’.

Line 281: Noted and changed.

Line 291: Noted and changed

Line 306: Deleted hyphen.

Line 311: Deleted hyphen.

Line 312: Removed ‘respectively’.

Line 336: OK, done as follows: “For this section, sharks discarded in poor physical condition (assumed unlikely to survive) included individuals that were discarded dead (see bars on Fig 6).

Line 346: Bar description added to figure title.

Line 363: Comma deleted.

Line 391: Deleted hyphen.

Line 404: We don’t see the difference

Line 405: Replaced ‘caused’ with ‘incurred’.

Line 408: Clarified that it was both study [10] and [24].

Line 415: Noted and changed.

Line 425: Added in comma.

Line 443: Changed to ‘depredating’.

Line 449: Noted and changed.

Line 454: Deleted hyphen.

Line 478: Noted and changed.

Line 487: Changed ‘fishermen’ to ‘fishers’.

---

## [Decision Letter · Decision Letter 1]

5 Aug 2020

PONE-D-20-12467R1

Shark discards in selective and mixed-species pelagic longline fisheries

PLOS ONE

Dear Dr. Jordaan,

Thank you for submitting your manuscript to PLOS ONE. After careful consideration, we feel that it has merit but does not fully meet PLOS ONE’s publication criteria as it currently stands. Therefore, we invite you to submit a revised version of the manuscript that addresses the points raised during the review process.

There are still concerns related to the quality of the available data. Namely, the use of only one year of data being not comprehensive of all quarters, not randomized, not equally distributed. The authors should provide further explanations on the above and make such limitations of their datasets more evident in the ms. The need for improved data collection schemes should also be highlighted.

We look forward to receiving your revised manuscript.

Kind regards,

Christos Maravelias, Ph.D.

Academic Editor

PLOS ONE

Reviewers' comments:

Reviewer's Responses to Questions

**Comments to the Author**

1. If the authors have adequately addressed your comments raised in a previous round of review and you feel that this manuscript is now acceptable for publication, you may indicate that here to bypass the “Comments to the Author” section, enter your conflict of interest statement in the “Confidential to Editor” section, and submit your "Accept" recommendation.

Reviewer #1: (No Response)

Reviewer #2: All comments have been addressed

2. Is the manuscript technically sound, and do the data support the conclusions?

Reviewer #1: Partly

Reviewer #2: (No Response)

3. Has the statistical analysis been performed appropriately and rigorously? 

Reviewer #1: No

Reviewer #2: (No Response)

4. Have the authors made all data underlying the findings in their manuscript fully available?

Reviewer #1: Yes

Reviewer #2: (No Response)

5. Is the manuscript presented in an intelligible fashion and written in standard English?

Reviewer #1: Yes

Reviewer #2: (No Response)

6. Review Comments to the Author

Reviewer #1: Concerns remain over the quality of the available data - only one year, not comprehensive of all quarters, not randomized, not equally distributed, etc. Improvements have been made in the manuscript by correcting errors in reported values, however, which reduced confusion regarding data exclusion. This study is of value to the field, and future work will benefit greatly from improved data collection efforts.

Regarding statistical analysis - seems as though observer data from 2018 is included that should not be, and so re-evaluation is needed. If this is not the case, the analysis is acceptable.

See specific comments in attached document.

Reviewer #2: (No Response)

7. PLOS authors have the option to publish the peer review history of their article (what does this mean?). If published, this will include your full peer review and any attached files.

Reviewer #1: No

Reviewer #2: No

---

## [Author Response · Author response to Decision Letter 1]

14 Aug 2020

REVIEWER 1 AND EDITORS COMMENTS

Overall comments

Reviewer 1: Concerns remain over the quality of the available data - only one year, not comprehensive of all quarters, not randomized, not equally distributed, etc. Improvements have been made in the manuscript by correcting errors in reported values, however, which reduced confusion regarding data exclusion. This study is of value to the field, and future work will benefit greatly from improved data collection efforts.

Editor: There are still concerns related to the quality of the available data. Namely, the use of only one year of data being not comprehensive of all quarters, not randomized, not equally distributed. The authors should provide further explanations on the above and make such limitations of their datasets more evident in the ms. The need for improved data collection schemes should also be highlighted.

Authors: We are fully aware of the quality of the data, which is unfortunately typical of data collected on commercial fishing vessels, where sampling opportunities are restricted by numerous factors. When and where a vessel goes depends on decisions taken by the fishing company, aiming to maximize catches and profits. Sailing dates are highly inconstant, depending on weather, demand for fish, vessel breakdowns or information on where fish are congregating. Vessels leave from different ports and sometimes don’t have space for an observer; at-sea decisions based on perceived fish abundance may lead to sudden changes in areas fished, or even changes in the species or sizes retained or discarded. We do not offer these as excuses – rather, they are simply the realities of fisheries-independent sampling on commercial fishing vessels. For these reasons, the data generated by fisheries-independent observers will very rarely be comprehensive of all quarters, equally distributed across fishing areas, or be completely randomised.

We did, however, use all the tools to our disposal to mitigate sampling effects. This included using a single observer on all trips (GJ, the first author of the manuscript) to collect data in a consistent way; covering as many vessels as logistically possibly to evaluate variability in discard practices; complete quarterly (seasonal) and areal (spatial) coverage of at least one subfleet (subfleet 1); undertaking a post-hoc sampling trip in 2018 to increase sample size for subfleet 2; and stressing throughout the manuscript that higher levels of observer sampling are required to increase confidence in discard ratio estimates. Indeed, in the Discussion, we even highlight the potential bias in estimates for subfleets 2-4, where sampling did not cover all quarters and areas, and recommend ways to improve estimates. We are very confident, however, in the methodology that we have developed, and were delighted to see that Reviewer 1 agrees that it will be very useful in future studies. 

We have made changes to the text to highlight the limitations of the data, and stress throughout the manuscript that data collection should be improved (see highlights in attached manuscript) 

Reviewer 1: Reviewer spent hundreds of sea days on commercial vessels, including longliners, frequently with a 90-100% sampling rate, and therefore fully appreciates (and respects) the realities of at-sea data collection.

Authors: We thank the reviewer for the clarification. In our case, the observer also undertook other tasks – such as biological sampling of the captured sharks, which prevented him from observing all the hooks that came on-board. Our sampling strategy relied on sub-sampling of sections of the line being hauled, interspersed with periods for biological sampling. Nevertheless, sampling covered 56 – 80% of hooks hauled on observed trips, which is still a very high percentage, and an acceptable subsampling coverage. 

LINE BY LINE COMMENTS (from marginal comments in pdf)

Line 32: We have changed to singular, and now use the more correct term ‘fisheries-independent observer’ throughout the manuscript.

Line 39: Added in “among other factors”.

Line 51: Replaced “and” with “therefore”.

Line 52: Deleted “therefore”.

Line 91: Added comma before “which”.

Line 145: We have edited the text (see track changes) to clarify the processing used on the different species – (i.e. removal of head, gills, gut and fins etc.) before weighing them.

Line 147: The vessels that caught large numbers of shortfin makos targeted them in a known congregation area (Agulhas region) addressed in the Discussion section and also elaborated on in Groeneveld et al. 2014. Vessels actively targeting sharks (shortfin makos, in this case) have been grouped as subfleet 4, based on their landings during 2015. We have adapted the paragraph (lines 500-502 in the revised manuscript with tracked changes) to include spatio-temporal closures for shortfin mako nursery areas and not only blue shark nursery areas as stated before.

Line 177: See response above (for line 32)

Table 1: Made corrections to total catch.

Line 235: See response above (for line 32). 

Line 269 to 270: Text added in the section Materials and Methods – Observer sampling to address the comments regarding addition of a sample in 2018 (See track changes). 

Line 327: Changed “importance” to “number”.

Line 332: Changed “confirmed” to “supported”.

Line 365: This is the correct number. Correction made in lines 460 of the revised manuscript with track changes.

Line 386: See comment above

Line 446 to 447: Corrected to 69 687.

Figure 5: The correction was made in Table 1.

---

## [Editor Report · Decision Letter 2]

20 Aug 2020

Shark discards in selective and mixed-species pelagic longline fisheries

PONE-D-20-12467R2

Dear Dr. Jordaan,

We’re pleased to inform you that your manuscript has been judged scientifically suitable for publication and will be formally accepted for publication once it meets all outstanding technical requirements.

Kind regards,

Christos Maravelias, Ph.D.

Academic Editor

PLOS ONE
---

## [Editor Report · Acceptance letter]

21 Aug 2020

PONE-D-20-12467R2 

Shark discards in selective and mixed-species pelagic longline fisheries 

Dear Dr. Jordaan:

I'm pleased to inform you that your manuscript has been deemed suitable for publication in PLOS ONE. Congratulations! Your manuscript is now with our production department. 

Kind regards, 

on behalf of

Dr. Christos Maravelias 

Academic Editor

PLOS ONE